# Type IV Collagen and SOX9 Are Molecular Targets of BET Inhibition in Experimental Glomerulosclerosis

**DOI:** 10.3390/ijms24010486

**Published:** 2022-12-28

**Authors:** José Luis Morgado-Pascual, Beatriz Suarez-Alvarez, Vanessa Marchant, Pamela Basantes, Pierre-Louis Tharaux, Alberto Ortiz, Carlos Lopez-Larrea, Marta Ruiz-Ortega, Sandra Rayego-Mateos

**Affiliations:** 1Cellular Biology in Renal Diseases Laboratory, Jiménez Díaz Foundation Health Research Institute, Autonomous University of Madrid, 28040 Madrid, Spain; 2Maimonides Biomedical Research Institute of Cordoba (IMIBIC), Cordoba University, 14004 Cordoba, Spain; 3REDINREN Spain/Ricord2040, 28029 Madrid, Spain; 4Translational Immunology, Principality of Asturias Health Research Institute (ISPA), Central University Hospital of Asturias, 33011 Oviedo, Spain; 5Paris Cardiovascular Center—PARCC, INSERM, Paris Cité University, 75015 Paris, France; 6Division of Nephrology and Hypertension, Jiménez Díaz Foundation Health Research Institute, Autonomous University of Madrid, 28040 Madrid, Spain

**Keywords:** glomerular diseases, BET, fibrosis, SOX9, epigenetic, kidney

## Abstract

Progressive glomerulonephritis (GN) is characterized by an excessive accumulation of extracellular (ECM) proteins, mainly type IV collagen (COLIV), in the glomerulus leading to glomerulosclerosis. The current therapeutic approach to GN is suboptimal. Epigenetic drugs could be novel therapeutic options for human disease. Among these drugs, bromodomain and extra-terminal domain (BET) inhibitors (iBETs) have shown beneficial effects in experimental kidney disease and fibrotic disorders. Sex-determining region Y-box 9 (SOX9) is a transcription factor involved in regulating proliferation, migration, and regeneration, but its role in kidney fibrosis is still unclear. We investigated whether iBETs could regulate ECM accumulation in experimental GN and evaluated the role of SOX9 in this process. For this purpose, we tested the iBET JQ1 in mice with anti-glomerular basement membrane nephritis induced by nephrotoxic serum (NTS). In NTS-injected mice, JQ1 treatment reduced glomerular ECM deposition, mainly by inhibiting glomerular COLIV accumulation and *Col4a3* gene overexpression. Moreover, chromatin immunoprecipitation assays demonstrated that JQ1 inhibited the recruitment and binding of BRD4 to the *Col4a3* promoter and reduced its transcription. Active SOX9 was found in the nuclei of glomerular cells of NTS-injured kidneys, mainly in COLIV-stained regions. JQ1 treatment blocked SOX9 nuclear translocation in injured kidneys. Moreover, in vitro JQ1 blocked TGF-β1-induced SOX9 activation and ECM production in cultured mesangial cells. Additionally, SOX9 gene silencing inhibited ECM production, including COLIV production. Our results demonstrated that JQ1 inhibited SOX9/COLIV, to reduce experimental glomerulosclerosis, supporting further research of iBET as a potential therapeutic option in progressive glomerulosclerosis.

## 1. Introduction

Chronic kidney disease (CKD) is predicted to become one of the top five causes of death in 2040. Although current therapeutic options slow CKD progression, many patients still progress to end-stage kidney disease (ESKD) [1]. In this context, SGLT2 inhibitors, initially used for diabetic kidney disease, have emerged as a potential treatment for non-diabetic CKD. However, most forms of glomerulonephritis (GN), including anti-basement glomerular membrane (GBM) nephritis, are immune-mediated and may cause a rapid decline in renal function [2,3]. Immunosuppressive therapy may induce remission, but many patients still have a high residual risk of ESKD [4], justifying the relevance of finding novel therapeutic strategies to prevent or slow disease progression. GN is characterized by excessive extracellular matrix (ECM) accumulation, mainly type IV collagen (COLIV), in the glomeruli, suggesting that antifibrotic drugs could inhibit glomerulosclerosis and disease progression. COLIV has particular relevance in human GN. In Goodpasture syndrome, autoantibodies target an epitope localized to the A3 NC1 domain of COLIV, and these autoantibodies accumulate in glomerular basement membranes [5,6]. Specific mutations in *ColIV* genes cause Alport syndrome [7,8,9,10,11] in an X-linked (*COL4A5*) an autosomal recessive, or an autosomal dominant inheritance (*COL4A3* or *COL4A4*) [7,12].

Epigenetic mechanisms contribute to kidney disease [13]. Epigenetic modulations are DNA or histone modifications responsive to environmental stimuli that regulate gene expression without altering the DNA sequence [14,15]. Epigenetic drugs have been proposed as therapeutic options, including in kidney diseases [15,16]. Among epigenetic drugs, inhibitors of protein–protein interactions of epigenetic readers are of special interest. The bromodomain (BRD) and extra-terminal (BET) protein family members (BRD2, BRD3, BRD4, and BRDT) are epigenetic readers. These proteins recognize acetylated lysine residues on proteins, including histones and transcription factors, regulating gene transcription [17,18]. The binding of BRD to acetyl–lysine residues on proteins can be blocked by selective inhibitors of BET proteins (iBETs), such as JQ1, I-BET151, GSK726, or CG223, that have been used as epigenetic therapeutic drugs [16,19,20,21,22]. Preclinical studies of iBETs have shown BDR involvement in many physiological and pathological processes, including autoimmunity, inflammation, adipogenesis, malignancy, infection and kidney damage [15,16,23,24,25,26,27].

Inhibitors of BET proteins have antifibrotic effects, as evidenced by reduced ECM production in preclinical pulmonary, kidney, and liver fibrosis [22,28,29,30,31]. The anti-fibrotic effects were mediated by the inhibition of fibroblast proliferation [28,29] or decreased inflammation [22,28,29]. In fibroblasts from patients with idiopathic pulmonary fibrosis, JQ1 reduced migration, proliferation, and IL6 release [28]. JQ1 also inhibited the activation of hepatic stellate cells and their differentiation into myofibroblasts modulating liver fibrosis [32]. Accordingly, in cultured kidney epithelial cells and fibroblasts iBETs restored phenotype changes induced by profibrotic factors and reduced experimental kidney fibrosis [31,33]. However, the direct effect of iBETs on ECM protein regulation in the context of GN is not well characterized.

Sex-determining region Y-box 9 (SOX9) is a transcription factor of the SOX family that has emerged as an important transcriptional regulator [34]. During development, activation of SOX9 is involved in testis, spinal cord, and cartilage formation, regulating collagen production and participating in kidney development [35,36,37,38,39,40,41]. SOX9 is upregulated in several tumor types and regulates cell growth, migration, invasion [42,43,44] and liver fibrosis [45,46]. In models of acute kidney injury (AKI), SOX9 activation precedes the expression of kidney damage biomarkers, such as NGAL and KIM-1, suggesting a very early role in kidney damage [47]. In ischemia–reperfusion AKI, SOX9 was activated in most proliferating cells. Furthermore, specific SOX9 deletion in proximal tubule cells reduced epithelial proliferation and increased injury severity [37,38]. These data suggest that SOX9 plays an essential role in tubular cell regeneration following AKI, but its role in progressive kidney fibrosis is not well established.

This study aimed to evaluate whether iBETs regulate ECM accumulation in experimental GN, evaluating the mechanisms involved, specifically, the role of SOX9. We have used a model of anti-GBM nephritis induced by nephrotoxic serum (NTS) administration in mice, an experimental model commonly used to study human GN [48,49]. We previously described that, in this model, treatment with the iBET JQ1 ameliorated kidney function and reduced glomerular lesions, mainly by the inhibition of NF-κB and NOTCH-signaling pathways activation and subsequent regulation of proinflammatory related genes [50,51], but whether BET inhibition could be directly involved in the regulation of glomerulosclerosis has not been investigated.

## 2. Results

### 2.1. BET Inhibition Reduced Glomerulosclerosis in Experimental Nephrotoxic Nephritis

The NTS model is characterized by glomerular ECM accumulation, mainly mediated by mesangial fibrosis and loss of podocytes, resulting in renal dysfunction and, therefore, resembling human GN [48]. We previously described how, in this model, JQ1 improved kidney function and reduced glomerular lesions [50]. Now, we confirmed these changes in kidney morphology, showing that JQ1 inhibited collagen accumulation and fibrinoid necrosis (Figure 1A,B).

### 2.2. BET Inhibition Reduced Glomerular Type IV Collagen in Experimental Glomerulonephritis

Next, changes in ECM composition were evaluated at the gene and protein levels. Real-time PCR showed upregulation of *fibronectin* gene levels in NTS kidneys, compared to healthy controls (Figure 2A). Nonetheless, no changes were found in other ECM components, like *type I collagen (ColI)* or profibrotic markers, such as *Ccn2* (Figure 2A). Notably, a marked upregulation of gene expression of *α3 chain type IV collagen* (*Col4a3*) and protein expression levels of type IV collagen (COLIV) were found in NTS-injured kidneys that were significantly reduced in JQ1-treated kidneys (Figure 2A,B). Moreover, an excess of COLIV deposition was detected by immunohistochemistry in the glomerulus of NTS-injured mice kidneys and was reduced by JQ1 (Figure 2C,D).

### 2.3. Type IV Collagen Is One of the Specific Targets of BET Inhibition

To further evaluate whether iBETs could specifically regulate *Col4a3* gene expression in injured kidneys, kidney chromatin immunoprecipitation (ChIP) assays were performed. ChIP showed increased BRD4 binding to the promoter region of *Col4a3* in NTS-injured kidneys compared to the controls, supporting direct regulation of *Col4a3* gene expression by BRD4. In addition, JQ1 reduced BRD4 enrichment (Figure 3), indicating that JQ1 inhibited the recruitment and binding of BRD4 to the *Col4a3* promoter region and, therefore, reduced *Col4a3* transcription, which clearly showed that *Col4a3* was a direct target of JQ1.

### 2.4. BET Inhibition Reduced ECM Production in Activated Mesangial Cells In Vitro

In pathologic conditions, glomerular mesangial cells are activated to increase cell proliferation, ECM deposition, secretion of inflammatory factors and the expression of adhesion molecules that participate in glomerulosclerosis [52]. Thus, we analyzed whether iBETs modulate ECM components in human mesangial cells (MCs; K18 cell line). MCs were stimulated with the profibrotic factor TGF-β1 for 48 h. Preincubation of MCs with JQ1 significantly decreased the mRNA expression of several ECM components, including *Col4a3*, *ColIa2* and *Fibronectin*, induced by TGF-β (Figure 4A). Moreover, JQ1 decreased the TGF-β1-stimulated release of fibronectin and COLIV into the extracellular medium (Figure 4B). The iBET JQ1 also modulated ECM production in renal fibroblasts, the primary ECM-producing cells. In cultured renal fibroblasts (TFB cell line), treatment with JQ1 also significantly reduced the TGF-β1-stimulated protein levels of fibronectin and COLI at 48 h (Figure 4C).

### 2.5. JQ1 Blocks SOX9 Activation in Experimental Nephrotoxic Nephritis

Previous gene expression studies using microarrays showed a high expression of the SOX9 gene in damaged glomeruli [53]. In NTS mice, renal SOX9 mRNA levels were upregulated, compared to control mice, and were reduced by JQ1 treatment (Figure 5A). Notably, nuclear SOX9 was increased in injured kidneys, reflecting the activation of this transcription factor (Figure 5B). Moreover, iBET treatment blocked renal SOX9 activation, as nuclear SOX9 levels were similar in JQ-1-treated NTS mice and control mice (Figure 5B).

### 2.6. JQ1 Blocks SOX9 Activation in Cultured Mesangial Cells

Mesangial cell stimulation with TGF-β1 for 24 h induced the translocation of SOX9 to the nucleus, where it can act as a transcription factor. In contrast, in cells treated with the iBET JQ1, SOX9 remained in the cytosol, thus preventing its activation (Figure 6A). Similar results were obtained when SOX9 protein levels were analyzed by Western blot. In nuclear extracts from mesangial cells stimulated with TGF-β1, JQ1 pretreatment reduced SOX9 protein levels to control values (Figure 6B).

### 2.7. SOX9 Regulates Type IV Collagen Production in Cultured Mesangial

To further determine the involvement of SOX9 in COLIV regulation in cultured human mesangial cells, the *SOX9* gene was silenced using a specific SOX9 siRNA. In SOX9 silenced cells, COLIV protein release induced by TGF-β1 was inhibited (Figure 7).

### 2.8. Activation of SOX9 and Increased Collagen IV in Experimental Nephrotoxic Nephritis

We further evaluated the potential relationship between SOX9 and COLIV in experimental GN. Using double confocal microscopy, nuclear SOX9 staining was co-localized with increased glomerular COLIV deposition in injured kidneys of NTS mice but was not observed in NTS mice treated with JQ1 (Figure 8).

### 2.9. JQ1 Inhibits the SMAD Pathway in Renal Cells and in Experimental Kidney Fibrosis

Activation of the TGF-β/SMAD pathway has been described, both in human and animal models of CKD [54]. Therefore, we evaluated whether iBETs modulated SMAD signaling in human mesangial cells and in experimental GN. In NTS mice kidneys, the SMAD pathway was activated, as shown by elevated renal levels of SMAD3 phosphorylation. In NTS mice treated with JQ1, renal phosphorylated SMAD3 levels markedly decreased (Figure 9A). In cultured mesangial cells, stimulation for 24 h with TGF-β1 increased SMAD3 phosphorylation, which was significantly decreased in cells preincubated with JQ1 (Figure 9B).

### 2.10. SOX9 Nuclear Translocation and Interstitial Fibrosis Are Inhibited by JQ1 in Experimental Renal Fibrosis Induced by Unilateral Ureteral Obstruction (UUO)

In murine UUO, we further evaluated whether iBETs regulated SOX9 and its potential relation to renal fibrosis. *Sox9* gene expression was significantly increased in obstructed kidneys, compared to control kidneys, and significantly decreased in obstructed kidneys from mice treated with JQ1 (Figure 10A). Nuclear protein levels of SOX9 were also markedly increased in obstructed kidneys and reduced by JQ1 treatment (Figure 10B). In addition, immunofluorescence revealed SOX9 nuclear localization in tubuloepithelial cells only in obstructed injured kidneys. Moreover, SOX9 co-localized with the expression of α-actin, a marker of activated fibroblasts. In obstructed kidneys from mice treated with JQ1, nuclear translocation of SOX9 and α-actin staining were lower (Figure 10C). Moreover, the mRNA expression of the profibrotic markers *Pai1* or *Tgf-β1* and of the ECM component *fibronectin* was increased in obstructed kidneys (Figure 10D) and this was also significantly decreased by JQ-1 (Figure 10D). Accordingly, the renal overexpression of COLI and fibronectin proteins observed in obstructed kidneys was decreased by JQ1 (Figure 10E).

## 3. Discussion

The main finding of this study was that the specific iBET JQ1 reduced glomerular COLIV accumulation in experimental GN, as shown in murine NTS–induced glomerular damage through the following different mechanisms: (1) inhibiting the binding of BET proteins to acetylated residues in the promoter regions of *Col4a3*, directly inhibiting *Col4a3* gene overexpression, and (2) blocking the activation of the transcription factors SOX9 and SMAD3, which are both involved in the regulation of ECM proteins. These results suggested that epigenetic drugs targeting BET proteins could be of therapeutic value in the fibrotic response in GN.

The ECM is composed of collagens, laminins, large glycoproteins (fibronectin), proteoglycans and matricellular proteins, forming a 3D structure involved in the regulation of cell–ECM interactions and cell phenotype changes [55]. However, specific ECM composition varies between renal structures. In this sense, COLIV is an essential component of the glomerular basement membrane [7,56], whereas COLI is found in interstitial spaces and is associated with tubulointerstitial fibrosis [57]. The initial cellular and molecular mechanisms involved in kidney damage can differ depending on the CKD’s etiology. However, a common feature of progressive CKD is an aberrant collagen accumulation caused by increased ECM gene expression, mainly COLI [57], and lower ECM degradation. Importantly, in progressive glomerular diseases, ECM proteins, mostly COLIV [58,59,60], accumulate within glomeruli, leading to glomerulosclerosis and loss of renal function, and patients progress to ESKD. Moreover, *ColIV* mutations cause glomerular disease [7,8,9,10,11]. Our experimental data in GN, induced by NTS, showed increased glomerular COLIV deposition, as well as upregulation of *Col4a3* gene expression, whereas COLI was not modified. Interestingly, treatment with the iBET JQ1 significantly reduced the glomerular *Col4a3* gene overexpression and protein accumulation. Our in vitro experiments in cultured mesangial cells, an important cell type implicated in ECM accumulation in glomerulosclerosis, showed that JQ1 inhibited the gene expression and the production of several ECM proteins induced by TGF-β1, including COLI and COLIV, demonstrating that the inhibition of BET proteins directly limited ECM protein expression. Similar results were previously described in tubuloepithelial cells stimulated with TGF-β1, where the inhibition of BRD4 function through gene silencing or JQ1 treatment decreased the expression of profibrotic genes, such as α-SMA and fibronectin [61]. These data suggest that iBETs could be used as antifibrotic drugs for progressive GN.

Several preclinical studies in different fibrotic disorders have also described the antifibrotic effects of iBETs [30,31,62,63], including experimental models of kidney damage [31,61,64]. However, the mechanisms involved were not fully explained. In UUO, treatment with iBET151 downregulated ECM-related genes associated with the inhibition of renal fibroblasts activation and macrophage infiltration [31]. Other authors have confirmed the effect of iBETs in decreasing interstitial fibrosis in murine UUO and rat hyperuricemia induced by a combination of adenine and potassium oxonate [61,64], as we also observed here in UUO mice treated with JQ1. Accordingly, in experimental hypertensive nephropathy, induced by angiotensin II administration in mice, JQ1 alleviated interstitial fibrosis and blocked epithelial-mesenchymal transition. The latter point was also demonstrated in cultured tubular epithelial cells [33]. Since iBETs also have anti-inflammatory properties [19], the described antifibrotic effects can be secondary to the inhibition of the initial inflammatory response that precedes collagen accumulation in CKD [65]. One mechanism of action of iBETs is the blockade of the interaction between BET proteins and acetylated histones or transcription factors [16], and subsequent regulation of the transcription of specific genes. In this sense, iBETs reduce proinflammatory gene expression inhibiting renal inflammatory cell infiltration, which is mediated, in some cases, by the modulation of the transcription factor NF-κB [16] or the Th17/IL17 immune response associated with human Th17 differentiation, and also inhibiting IL17, IL21, and GMCSF expression [50,66,67]. Transcriptomics analyses in preclinical cardiac damage and cultured cells have described that iBETs reduced gene expression of innate inflammatory and profibrotic myocardial pathways related to NF-κB and TGF-β signaling [68]. In this sense, we previously found that, in NTS mice, there was neither upregulation of *Tgf-β1* gene expression nor any effect from JQ1 [51], suggesting that the beneficial effects of iBETs on ECM regulation in this mice model were independent of TGF-β1 signaling. Notably, the novel data presented here demonstrate that the iBET JQ1 could modulate glomerular fibrosis by directly targeting the COLIV genes. In NTS-induced GN, treatment with the iBET JQ1 significantly reduced renal *Col4a3* gene overexpression and decreased glomerular COLIV protein synthesis and accumulation. Using chromatin immunoprecipitation assays, we demonstrated that JQ1 modulated BRD4 binding to the promoter region of *Col4a3*, thus reducing *Col4a3* transcription, and identifying this gene as a specific target of BET inhibition.

The transcription factor, SOX9, plays an extensive role in the development of several organs, including the kidney, where it is involved in early nephron and ureter formation [69,70]. Under normal adult conditions, there is no renal expression of SOX9. Experimental studies suggest that the re-expression of SOX9 contributed to renal tubular regeneration after exposure to carcinogens or hydrogen peroxide [71]. SOX9-positive proximal tubular epithelial cells proliferate, expand, and differentiate to replace damaged tissue in acute kidney injury [37,38]. SOX9 is also involved in fibrotic disorders, in which it is a key transcriptional regulator of genes related to ECM formation [72]. In vivo loss of SOX9 attenuated cardiac fibrosis in response to ischemic injury [73,74], whereas sustained SOX9 expression was associated with experimental fibrosis [75,76]. SOX9 overexpression in cultured rat fibroblasts is involved in renal tubular EMT and ECM aggregation via the PI3K/AKT signaling pathway [76]. A previous study in murine immune GN induced by NTS in CD1ICR mice showed that SOX9, *Tgf-β1*, and *Col4a2* were dramatically increased [77]. In the same report, SOX9 blockade modulated *Col4a2* expression induced by TGF-β1 stimulation in cultured mesangial cells [77]. In contrast, no prior studies have addressed the link between SOX9 and *Col4a3* in the fibrotic response in GN. In our study, we confirmed that nuclear SOX9 levels were also increased in murine NTS and preferentially colocalized with glomerular COLIV positive cells. Moreover, increased *Col4a3* gene expression levels were found in NTS-injected mice. Importantly, JQ1 prevented SOX9 and *Col4a3* gene upregulation. Previous studies reported that, in the UUO model, SOX9 expression increased rapidly and peaked at 24 to 48 h, remaining elevated for several weeks after injury [38]. Now, we have found that nuclear expression of SOX9 is increased in obstructed kidneys, preferentially distributed in areas of tubulointerstitial fibrosis, and associated with cells expressing α-SMA, a marker of activated myofibroblasts (the main collagen-producing cells). Importantly, in mice treated with JQ1, neither nuclear SOX9 activation nor fibrosis was observed. In addition, in cultured mesangial cells, SOX9 gene silencing inhibited COLIV production induced by TGF-β1, demonstrating that this transcription factor directly regulated *ColIV* gene transcription. A recent report determined that the BRD4 inhibitor JQ1 controlled SOX9 expression via multiple regulatory mechanisms, including transcription regulation, BRD4-SOX9 protein-protein interaction, and further protein stability [78]. Our data describe, for the first time, how the iBET JQ1 inhibits the SOX9/COLIV signaling pathway in experimental GN and in cultured renal cells, identifying a novel mechanism involved in the beneficial effects of iBETs in renal fibrosis.

The SMAD signaling pathway is one of the main regulatory mechanisms in the fibrotic response [79,80,81]. A previous study described how BET inhibitors (I-BET151) also blocked SMAD activation in the UUO model in mice [31]. In human mesangial cells, we found that JQ1 inhibited SMAD3 activation induced by TGF-β1, as shown by decreased SMAD3 phosphorylation levels. Accordingly, the SMAD pathway was inhibited in NTS mice treated with JQ1. Several studies have confirmed that JQ1 attenuated fibrosis and inflammation resulting from radiation used in radiotherapy in thoracic cancer by suppressing BRD4, c-MYC, p65 NF-κB, and the COLII/TGF-β/SMAD pathway [29]. Indeed, in liver injury, iBET blocked TGFβ/SMAD signaling to inactivate hepatic stellate cells [82]. Moreover, in the umbilical vein, endothelial cells and in mouse aortic endothelial cells, both the pharmacological inhibition of BRD4 with JQ1 or BRD4 silencing, and the downregulation of SMAD signaling, blocked endothelial to mesenchymal transition, migration, and ECM synthesis induced by TGF-β1 [83]. Regulation of TGF-β signaling is highly complex and can be manipulated at multiple levels. Interestingly, TGF-β transcriptional activity can be alter by acetylation/de-acetylation of specific lysine residues in Smad2/3 [84,85], and can, therefore, be a target of BETi. Moreover, BRD4 can directly regulate profibrotic genes, as demonstrated by cistromic analyses, revealing that BRD4 is highly enriched at enhancers colocalized with profibrotic transcription factors [86]. Our data suggest that BET proteins modulated the SMAD pathway in experimental renal fibrosis, but future studies are needed to unravel the mechanism involved in SMAD pathway regulation.

Our analyses of the role of a selective iBET, such as JQ1, focused on the role of bromodomain/acetyl–lysine binding as a principal mechanism in the modulation of chromatin biology and gene transcription and may have translational value. Previous studies described the possible toxicity of high doses of JQ1 (100–300 mg/kg/day) in mice that could result in body weight loss and have an effect on lymphoid and hematopoietic cell integrity [87], but in our study we did not observe, in mice treated with JQ1, any change in animal body weight or histological damage associated with possible toxicity. Less toxic iBETs, such as ABBV-075 (BRD2/4/T pan inhibitor) and Apabetalone/RVX-208 (BRD2 specific inhibitor), were developed and tested in humans [16]. There are several clinical reports about the role of iBETs in malignancy (ODM-207: NCT03035591 and ABBV-075: [83]); including leukemia and multiple myeloma (OTX015: NCT01713582; RO6870810: NCT02308761 and CPI-0610: NCT02157636), resistant prostate cancer (ZEN-3694; [84]); non-Hodgkin’s lymphoma (CC-90010: NCT03220347); and diffuse large B-cell lymphoma (RO6870810: NCT01987362). In the cardiovascular field, iBETs also have beneficial effects, as shown by studies in atherosclerosis (ASSURE Trial/RVX-208/Apabetalone: NCT01067820) and acute coronary syndrome (BETonMACE trial/ RVX-208/Apabetalone: NCT02586155). In kidney disease, a phase II study in Fabry disease described the role of RVX000222 in cardiovascular damage (NCT03228940).

In conclusion, JQ1 reduced experimental glomerulosclerosis by inhibiting SOX9/COLIV. These results suggest that BET inhibitors could have important therapeutic applications in CKD, including in progressive glomerulosclerosis.

## 4. Materials and Methods

### 4.1. Ethics Statement

All animal procedures were performed in 3-month-old male C57BL/6 mice, according to the guidelines of animal research in the European Community and with prior approval by the Animal Ethics Committee of the Health Research Institute IIS-Fundación Jiménez Díaz.

### 4.2. Experimental Models

The BET bromodomain inhibitor JQ1, a thieno-triazolo-1,4-diazepine, was synthesized and provided collaboratively by Dr. James Bradner (Dana-Farber Cancer Institute, Boston, MA, USA) [85]. For in vivo studies, JQ1 was dissolved in 10% hydroxypropyl β-cyclodextrin and used at a therapeutic dose (100 mg/kg/day, i.p.), as previously described [32]. No signs of toxicity and damage were observed in JQ1 administrated animals.

Anti-murine glomerular basement membrane (GBM) nephritis was induced in male C57BL/6 mice by administering rabbit nephrotoxic serum (NTS) by intravenous injection as described [30]: 50 μL of NTS diluted 1/10 in sterile saline on day 1; then, 4 μL/g body weight on days 2 and 3, and mice were studied 10 days later. Four groups of mice were studied: NTS (*n* = 7) and NTS treated with JQ1 (*n* = 7), vehicle controls (*n* = 5). JQ1, or vehicle, were started one day before the first NTS administration and then administered daily.

Unilateral ureteral obstruction (UUO) was induced in male C57BL/6 mice by surgical ligation of one ureter under isoflurane anesthesia, as previously described [87]. Some animals were treated daily with JQ1 from 1 day before UUO and studied after 5 days (*n* = 6–7 mice per group).

Animals were killed with 5 mg/kg xylazine (Rompun, Bayer AG) and 35 mg/kg ketamine (Ketolar, Pfizer), and the kidneys were perfused with cold saline before removal and processed for immunohistochemistry (fixed and paraffin-embedded tissue), immunofluorescence (OCT frozen tissue), ChIP assays (fixed in 1% formaldehyde followed by quenching with 0.125 M glycine), or for RNA and protein studies (snap frozen in liquid nitrogen).

### 4.3. In Vitro Studies

The K18 cells (immortalized mesangial human cell line) were grown in DMEM with 10% FBS, 100 U/mL penicillin, and 100 μg/mL streptomycin at 37 °C in 5% CO_2_. Subconfluent cells (60,000 cells/cm^2^) were incubated with stimuli in serum-free medium or with 1% FBS (fibroblasts TFB cells) for 15 min or 48 h. JQ1 was resuspended in DMSO (0.25%). Cells were rested in serum-free medium for 24–48 h and were pretreated with 5 uM JQ1 for 1 h and stimulated with 10 ng/mL of TGF-β1 for different periods.

### 4.4. Histology and Immunohistochemistry

Paraffin-embedded kidney sections were stained using standard histology procedures as described elsewhere [88]. Kidney injury was evaluated by Masson staining. Extracapillary proliferation ratio was calculated by counting injured and normal glomeruli. The proportion of pathological glomeruli was assessed by examination of at least 50 glomeruli per section by an examiner masked to the experimental conditions.

Immunostaining was performed in 3 μm thick tissue sections. Antigen retrieval was performed using the PTlink system (Dako; Santa Clara, CA, USA) with sodium citrate buffer (10 mM) adjusted to pH 6–9, depending on the immunohistochemical marker. Endogenous peroxidase was blocked. Tissue sections were incubated for 1 h at room temperature with 4% BSA and 10% of a specific serum (depending on the secondary antibody used) in PBS to eliminate non-specific protein binding sites. Primary antibodies were incubated overnight at 4 °C. Specific biotinylated secondary antibodies (Amersham Biosciences; Chalfont Buckinghamshire, UK) were used, followed by streptavidin–horseradish peroxidase conjugate and 3,3-diaminobenzidine as a chromogen. The primary antibody used was anti-type IV collagen (1/200; ABCAM). The omission of primary antibodies checked specificity. Quantification was made by determining in five to ten randomly chosen fields (×200 magnification) the total number of positive cells using Image-Pro Plus software (data expressed as the positive-stained area relative to the whole area) or by quantifying the number of positive nuclei manually.

### 4.5. Gene Expression Studies

RNA from cells or renal tissue was isolated with TRItidy G^TM^ (PanReac; Barcelona, Spain). cDNA was synthesized by a High-Capacity cDNA Archive kit (Applied Biosystems; Waltham, MA, USA), using 2 μg total RNA primed with random hexamer primers. Next, quantitative gene expression analysis was performed by real-time PCR on an AB7500 fast real-time PCR system (Applied Biosystems) using fluorogenic TaqMan MGB probes and primers designed by Assay-on-Demand^TM^ gene expression products. Human assay IDs were: Fibronectin Hs01549976_m1 and Type I Collagen Hs00164099_m1. Mouse assays IDs were: TGF-β1, Mm01178820_m1; serpine-1 (Pai-1) Mm00435858_m1, Fibronectin Mm01256744_m1, Type I Collagen Mm00483888_m1; SOX9 Mm00448840_m1 and Collagen 4a3 Mm00483669_m1. Data were normalized to *Gapdh*: Mm99999915_g1. The mRNA copy numbers were calculated for each sample by the instrument software using Ct value (“arithmetic fit point analysis for the lightcycler”). Results were expressed in copy numbers, calculated relative to unstimulated cells.

### 4.6. Protein Studies

Total protein samples for frozen renal tissue were isolated in lysis buffer (50 mmol/L Tris-HCl, 150 mol/L NaCl, 2 mmol/L EDTA, 2 mmol/L EGTA, 0.2% Triton X-100, 0.3% IGEPAL, 10 μL/mL proteinase inhibitor cocktail, 0.2 mmol/L PMSF, and 0.2 mmol/L orthovanadate) as described in [88]. In addition, nuclear and cytoplasmic fractions were separated from renal tissues or cells using the NE-PER Reagent (Pierce), following the manufacturer’s instructions. Proteins (20–100 μg per lane, quantified using a BCA protein assay kit) were separated on 8–12% polyacrylamide–SDS gels under reducing conditions [32]. For Western blotting, cell (50 μg/lane) protein extracts were separated on 8–12% polyacrylamide-SDS gels under reducing conditions. Samples were then transferred onto polyvinylidene difluoride membranes (Thermo Scientific), blocked with TBS/5% non-fat milk/0.05% Tween-20, and incubated overnight at 4 °C with the following antibodies (dilution): anti-SOX9 (1:500; Millipore; Burlington, MA, USA), anti-fibronectin (1/5000; Millipore); anti-type I collagen (1/5000; Millipore); anti-type IV collagen (1:1000; Abcam; Cambridge, UK), anti-type I collagen (1:1000; Millipore), and anti-p-Smad3 (1:1000; Abcam). Membranes were subsequently incubated with peroxidase-conjugated IgG secondary antibody and developed using an ECL chemiluminescence kit (Amersham Biosciences; Chalfont Buckinghamshire, UK). Loading controls were performed using an anti-GAPDH antibody (1:5000; CB1001, Millipore). Results were analyzed by LAS 4000 and Amersham Imager 600 (GEHealthcare Chicago, IL, USA) and densitometered by Quantity One software (Biorad).

### 4.7. Immunofluorescence Staining of Cells

For immunocytochemistry experiments, cells were grown on coverslips. After incubation, cells were fixed in paraformaldehyde 4% and permeabilized with 0.2% Triton-X100 for 10 min. After blocking with 3% BSA for 1 h, they were incubated with anti-SOX-9 ([1:100]; Millipore) overnight at 4 °C. Then, they were incubated for 1 h with the secondary antibody AlexaFluor^®^ 488 conjugated rabbit anti-mouse secondary antibody ([1/300]; Invitrogen; Waltham, MA, USA). Nuclei were stained with 4′,6-Diamidino-2-phenyindole (DAPI, Sigma-Aldrich; Saint Louis, MO, USA) as a control for equal cell density. The absence of a primary antibody was used as a negative control. Samples were mounted in Mowiol 40–88 (Sigma-Aldrich) and examined by a Leica TCS SP5 confocal microscope.

### 4.8. Immunofluorescence Staining of Tissue Samples

In paraffin-embedded kidney sections (3 µm), antigens were retrieved by the PTlink link system. After the slides were blocked with 10% BSA and 10% FBS for 1 h, they were incubated with anti-SOX9 (1/200; Millipore) or anti-type IV collagen (1:1000; Abcam) for 1 h, followed by an AlexaFluor^®^ 488 conjugated rabbit anti-mouse secondary antibody (1/200; Invitrogen); AlexaFluor^®^ 633 conjugated rabbit anti-mouse secondary antibody (1/200; Invitrogen) or AlexaFluor^®^ 568 conjugated rabbit anti-mouse secondary antibody (1/200; Invitrogen) for 1 h. The absence of a primary antibody was used as a negative control. Samples were mounted in prolong gold (Thermo Fisher Scientific; Waltham, MA, USA) and examined using a Leica DM-IRB confocal microscope.

### 4.9. Gene Silencing

Gene silencing in cultured cells was performed using either pre-designed siRNA corresponding to SOX-9 or their corresponding scramble siRNAs (Ambion; Austin, TX, USA). Subconfluent cells were transfected for 24 h with 25 nmol/L siRNA using 50 nmol/L Lipofectamine RNAiMAX (Invitrogen) or treated only with lipofectamine vehicle, according to the manufacturer’s instructions. The cells were then incubated with 10% heat-inactivated FBS for 24 h, followed by an additional 24 h in serum-free medium before the experiments.

### 4.10. Chromatin Immunoprecipitation

Kidneys were fixed in 1% formaldehyde (Sigma-Aldrich), followed by quenching with 0.125M glycine. DNA fragments 500–1000 bp long were generated on a BioRuptor (Diagenode; Denville, NJ, USA), and ChIP assays were performed using the High Cell ChIP kit (Diagenode) following the manufacturer’s instructions. The antibodies used were anti-BRD4 (Bethyl Laboratories; Montgomery, TX, USA) and normal IgG as negative control (Millipore). Immunoprecipitated DNA was analyzed by quantitative RT-PCR using the *Col4a3* primers: Forward: 5′-CCACTTCTCCCCTCCCTTAG-3′//Reverse 5′-CCCGGTGTTTTCTGTGTTCT -3′. Chromatin obtained before immunoprecipitation was used as the input control. Relative enrichment was calculated as the percentage of input DNA for each sample using the formula % input = 2 exp [(Ct unbound) − log_2_ (unbound dilution factor) − Ct bound)] × 100 and normalized to normal rabbit IgG antibody (considered as 1).

### 4.11. Statistical Analysis

Results are expressed as mean ± SEM of the n-fold increase with respect to the control (represented as 1). In the NTS nephritis model, data were obtained normalizing NTS and NTS + JQ1 kidneys versus control average. The Shapiro–Wilk test was used to evaluate sample normality distribution. If the samples followed the Gaussian distribution, a one-way ANOVA, followed by the corresponding post-hoc analyses of Fisher’s LSD test, was used. To compare non-parametric samples, a Kruskal–Wallis and subsequent post-hoc analysis of Uncorrected Dunn’s test was performed. Statistical analysis was conducted using GraphPad Prism 8.0 (GraphPad Software, San Diego, CA, USA). Values of *p* < 0.05 were considered statistically significant.

## Figures and Tables

**Figure 1 ijms-24-00486-f001:**
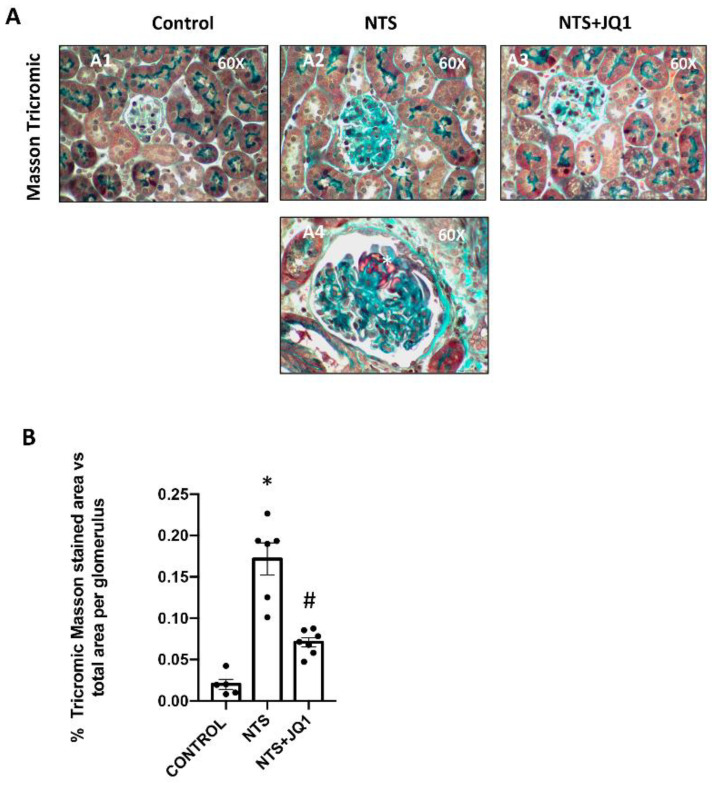
(**A**) JQ1 reduces glomerulosclerosis in nephrotoxic nephritis. Glomerulonephritis was induced in C57Bl/6 mice by the administration of NTS (A2, A3, A4), and mice were studied 10 days later. Mice were treated daily with JQ1 (100 mg/kg/day) (A3) or vehicle (A1), starting before the first NTS injection. In paraffin-embedded kidney sections, kidney morphology was evaluated by Masson Trichromic staining, and showed increased collagen content (A2) and fibrinoid necrosis (A4) in NTS mice. JQ1 diminished glomerular collagen deposition and severe glomerular lesions (A3), including fibrinoid necrosis (*). Panels show a representative animal from each group (600× magnification) focused on the glomerulus. (**B**) Masson Trichromic staining quantification. Scale bar represents 50 μm. * *p* < 0.05 vs. control; # *p* < 0.05 vs. NTS-injected mice.

**Figure 2 ijms-24-00486-f002:**
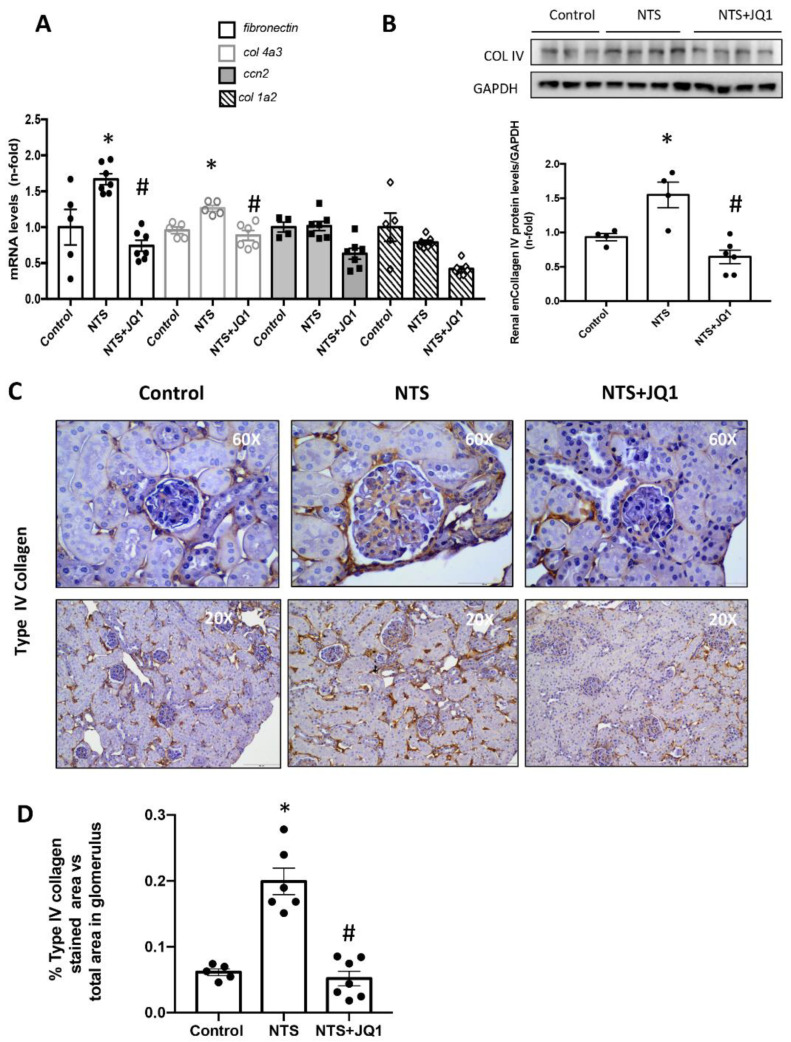
JQ1 reduced renal type IV collagen overexpression at the gene and protein levels in nephrotoxic nephritis. Mice were treated daily with JQ1 (100 mg/kg/day) or vehicle, starting before the first NTS injection. (**A**) RNA was isolated from frozen whole kidney samples, and α3 chain type IV collagen (*Col4a3*); type I collagen (*ColI*); *fibronectin*, and *Ccn2* gene expression levels were evaluated by real-time qPCR. (**B**) Protein levels of COLIV were quantified by Western blot in total kidney extracts. The figure shows representative blots where GAPDH was used as a loading control. Data are expressed as the mean ± SEM of 5–7 animals per group. * *p* < 0.05 vs. control; # *p* < 0.05 vs. NTS-injected mice. (**C**) Immunohistochemistry with an anti-type IV collagen antibody in paraffin-embedded kidney sections. A representative animal from each group was shown (200× magnification). A detailed image of glomeruli was included (600× magnification). (**D**) Type IV collagen staining quantification. Scale bar represents 100 μm (200×) and 50 μm (600×).

**Figure 3 ijms-24-00486-f003:**
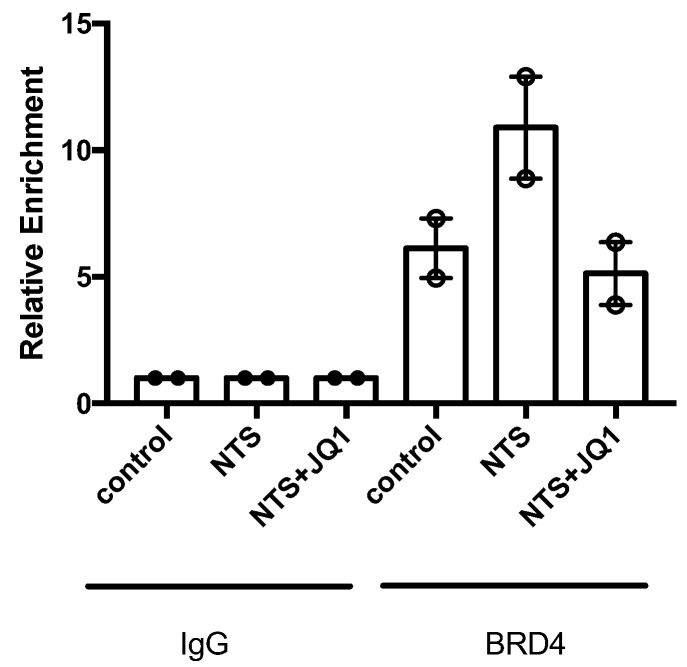
Type IV Collagen is a direct target of iBETs. ChIP assays were performed in renal samples using an antibody specific for BRD4 or normal rabbit IgG as a negative control. Enrichment of BRD4-binding regions in the promoter of mouse *Col4a3* was quantified by qPCR using specific primers. In each group, samples were pooled (4–7 mice per group). Data are from two independent experiments, and each qPCR was run in triplicate. Results are expressed as the n-fold enrichment of anti-BRD4 antibody relative to the negative control antibody (considered to be 1).

**Figure 4 ijms-24-00486-f004:**
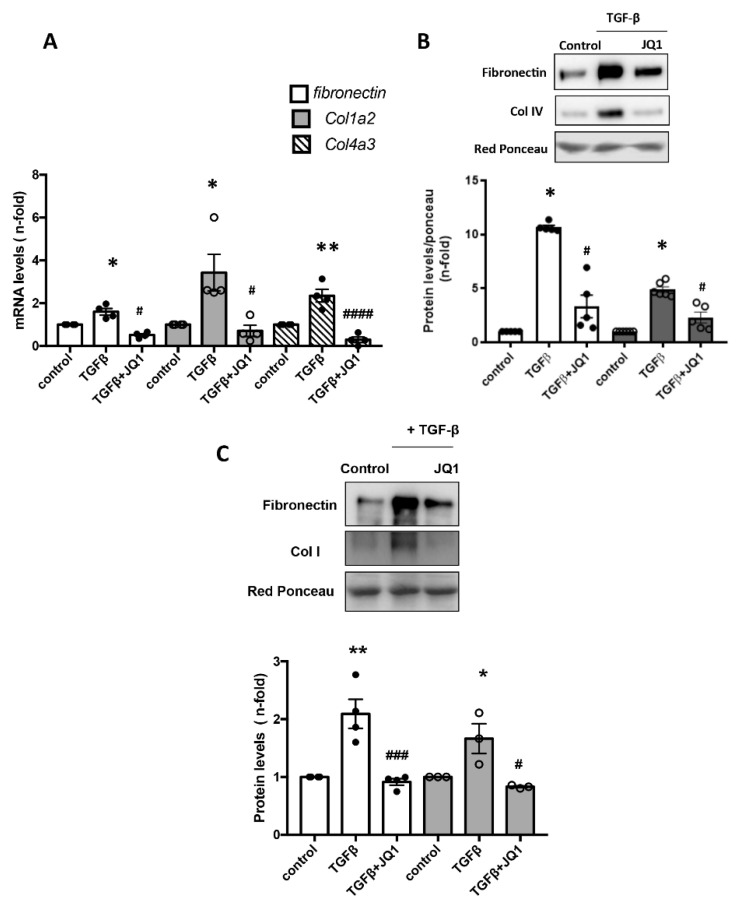
BET inhibition by JQ1 reduced ECM gene and protein production in human mesangial cells and murine kidney fibroblasts. Cells were pretreated with 5 uM JQ1 for 1 h and stimulated with 10 ng/mL TGF-β1 for 24 h (gene expression) or 48 h (protein expression). (**A**) *Fibronectin*, *Col4a3* and *ColIa2* gene levels were evaluated by real-time qPCR in mesangial cells. (**B**) Fibronectin and Type IV collagen (COLIV) protein levels were evaluated by Western blot in mesangial cells. (**C**) Protein levels of Fibronectin and Type I collagen (COLI) were evaluated by Western blot in kidney fibroblasts. * *p* < 0.05 vs. control; ** *p* < 0.001 vs. control; # *p* < 0.05 vs. treated cells with TGF-β1; ### *p* = 0.0001 vs. TGF-β1; #### *p* < 0.0001 vs. TGF-β1.

**Figure 5 ijms-24-00486-f005:**
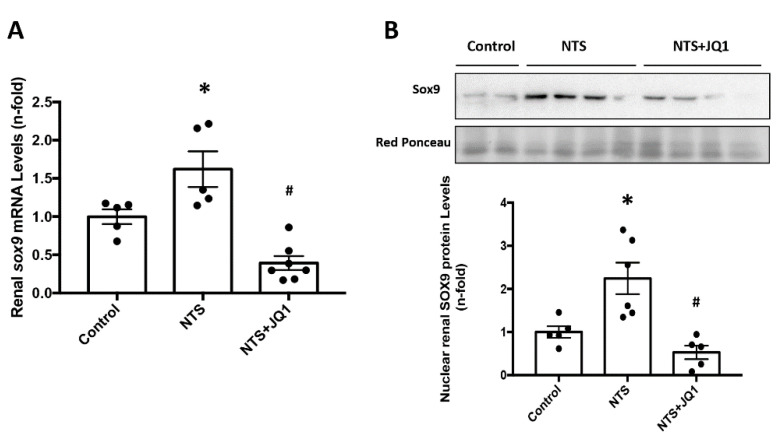
JQ1 reduces renal *Sox-9* gene overexpression and blocks SOX-9 nuclear translocation in experimental nephrotoxic nephritis. Mice were treated daily with JQ1 (100 mg/kg/day) or vehicle for 10 days, starting before the first NTS injection. (**A**) RNA was isolated from frozen whole kidney samples, and *Sox-9* gene expression levels were evaluated by real-time qPCR. (**B**) Protein levels of SOX9 were quantified by Western blot in nuclear kidney extracts. The figure shows representative blots where Red ponceau was used as a loading control. Data are expressed as the mean ± SEM of 5–7 animals per group. * *p* < 0.05 vs. control; # *p* < 0.05 vs. NTS-injected mice.

**Figure 6 ijms-24-00486-f006:**
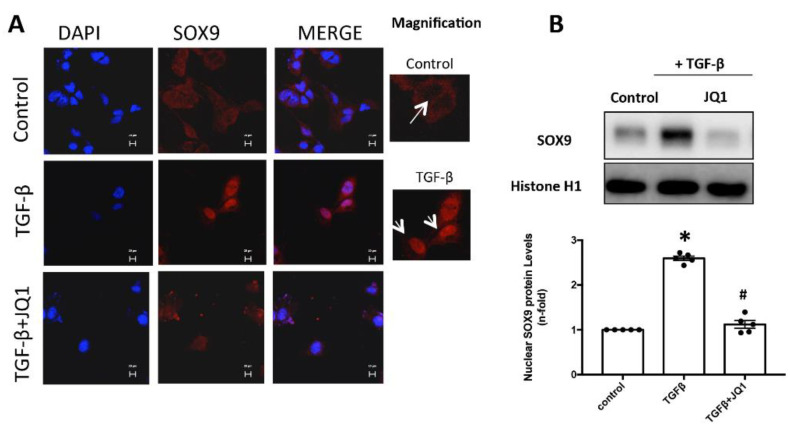
JQ1 blocked the nuclear translocation of SOX9 in human mesangial cells. Cells were pretreated with 5 uM JQ1 for 1 h and then stimulated with 10 ng/mL TGF-β1 for 24 h. The nuclear localization of SOX9 was evaluated by immunofluorescence using a secondary Alexa 688 antibody (**A**) and Western blot (**B**). Data are expressed as mean ± SEM of 3 independent experiments and scale bar represents 20 μM. * *p* < 0.05 vs. control # *p* < 0.05 vs. treated cells with TGF-β1. Arrows indicate the presence or absence of SOX9 nuclear translocation.

**Figure 7 ijms-24-00486-f007:**
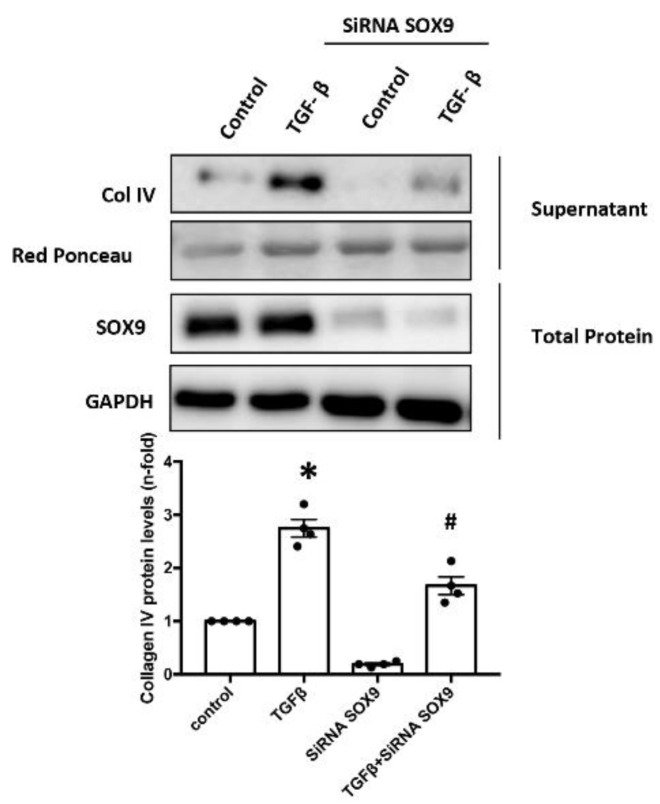
SOX9 gene silencing by specific siRNA decreases Type IV collagen protein levels in human mesangial cells. SOX9-silenced cells were stimulated with 10 ng/mL TGF-β1 for 48 h. COLIV protein levels were evaluated by Western blot in the supernatant of mesangial cells. Sox9 protein levels were evaluated by Western blot to corroborate a correct silencing protocol. Data are expressed as mean ± SEM of 3 independent experiments. * *p* < 0.05 vs. control # *p* < 0.05 vs. treated cells with TGF-β1.

**Figure 8 ijms-24-00486-f008:**
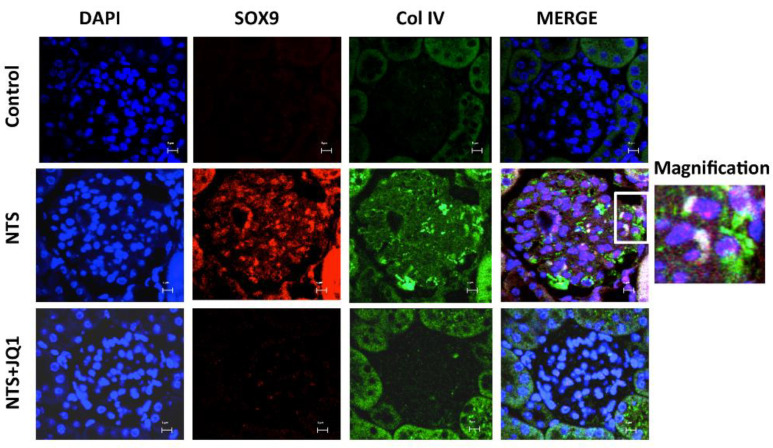
JQ1 treatment inhibits the expression and nuclear localization of SOX9 and the expression of type IV collagen in murine immune glomerulonephritis. Dual immunofluorescence staining for SOX9 and COLIV with two different fluorescent secondary antibodies, Alexa 568 and Alexa 488, respectively, disclosed nuclear translocation of SOX9 in NTS-induced glomerulonephritis, which colocalized with COLIV staining, and both were inhibited by JQ1. Scale bar represents 5 μM.

**Figure 9 ijms-24-00486-f009:**
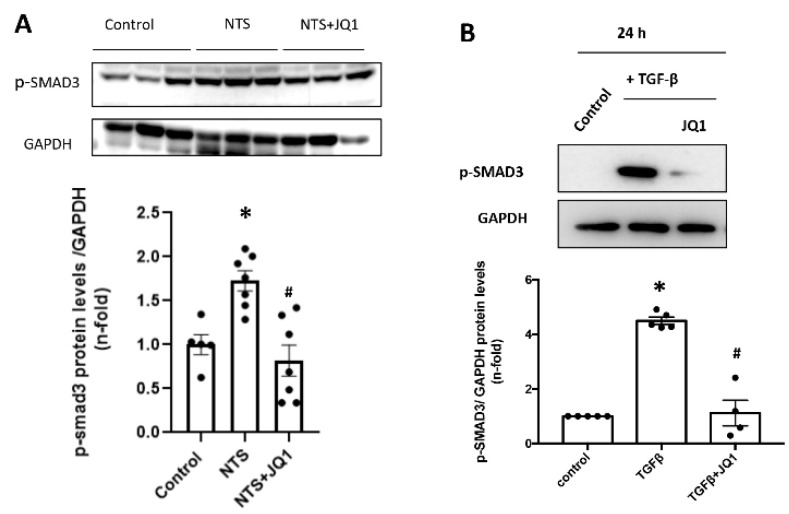
Inhibition of BET proteins decreased renal SMAD3 activation in NTS-induced nephritis and cultured mesangial cells. (**A**) Mice were treated daily with JQ1 (100 mg/kg/day) or vehicle for 10 days, starting before the first NTS injection, and total protein renal extracts were used. Data are expressed as the mean ± SEM of 6–8 animals per group. * *p* < 0.05 vs. control; # *p* < 0.05 vs. NTS. (**B**) K18 mesangial cells were pretreated with 5 uM JQ1 for 1 h and stimulated with 10 ng/mL TGF-β for 24 h. Protein levels of phosphorylated-SMAD3 (p-SMAD3) were evaluated by Western blot. Data are expressed as mean ± SEM of 3 independent experiments. * *p* < 0.05 vs. control # *p* < 0.05 vs. treated cells with TGF-β1.

**Figure 10 ijms-24-00486-f010:**
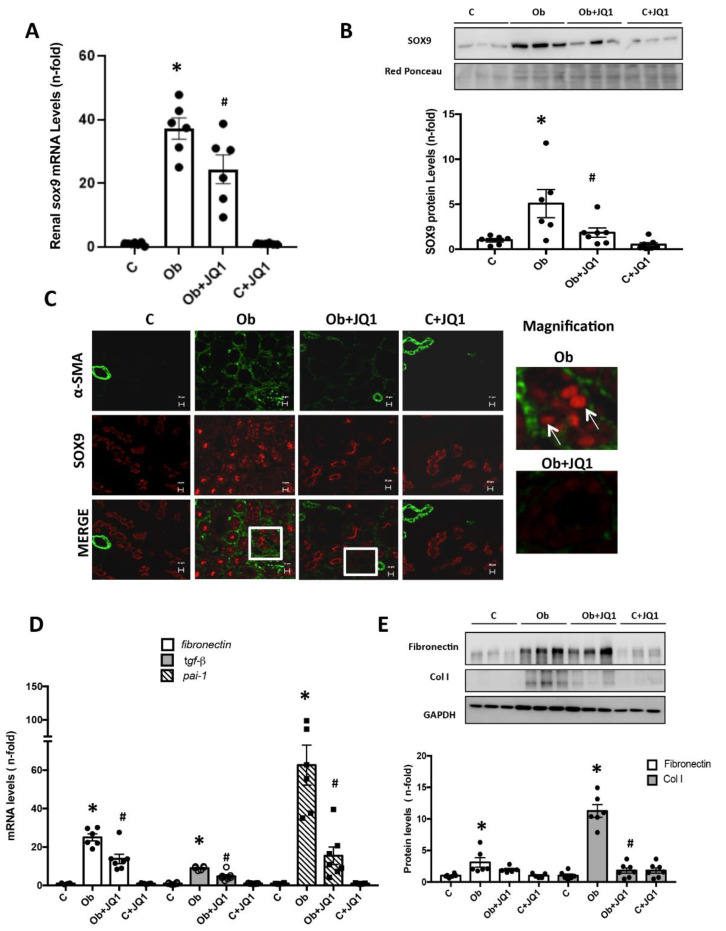
Inhibition of BET proteins by JQ1 prevents kidney SOX9 and ECM overexpression induced by UUO. (**A**,**B**) Gene and protein expression levels of SOX9 were evaluated by real-time qPCR (**A**) and Western blot (**B**). (**C**) The nuclear localization of SOX9 and its relationship with the extracellular matrix was evaluated by immunofluorescence. D, E) Inhibition of BET proteins by JQ1 decreases profibrotic markers in murine UUO such as (**D**) *Fibronectin*, *Pai-1*, and *Tgf-β1*, evaluated by real-time qPCR, or (**E**) The protein levels of Fibronectin and Type I Collagen (COLI) determined by Western Blot. Data are expressed as the mean ± SEM of 6–8 animals per group and scale bar represents 20 μM. * *p* < 0.05 vs. contralateral; # *p* < 0.05 vs. obstructed kidneys treated with JQ1.

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
