# Peer review of "Type IV Collagen and SOX9 Are Molecular Targets of BET Inhibition in Experimental Glomerulosclerosis"

_ijms, 2022, doi:10.3390/ijms24010486_

Round 1

Reviewer 1 Report

The manuscript by Margado – Pascual et al describes an interesting finding that inhibition of bromodomain and extra terminal domain proteins by selective drugs (iBET) can partially rescue glomerular injury due to excessive Col IV and other ECM protein secretion. The manuscript has 10 figures, they have tested the effect of iBET (JQ1) on animal and cell culture models of glomerular injury/disease. The experiments use nephrotoxic serum in animal model with simultaneous iBET (JQ1) treatment, mesangial cells induced with TGF-beta and combined treatment with JQ1, as well as an UUO model for kidney injury showing the effects of JQ1. There are no major issues in the manuscript related to conceptual and technical aspects of the experiments. Here are a few comments.

1)    It will be good to provide a few gels for the qPCR data. qPCR products can be run in gel to show the changes in levels for major genes like ColIV and Sox9.

2)    The ChiP data is represented only as bar graphs. No supplementary data is available to see the confirmation of the pulldown of Sox9 or the qPCR gel to provide more evidence. This is necessary to increase the confidence in this data.

3)    Supplemental western blot images provided lack ladder lanes? Why are the ladder lanes cut off? Please provide those.

4)    Discussion needs to include details about potential iBET toxicity and if the animal models handled the drug well and did not lead to any complications.

5)    Title and conclusion are good and correlates with the findings. The manuscript does not have functional evidence (kidney function tests) to suggest recovery from nephrotic injury, only that iBET decreases SOX9 and ColIV in their models.

Author Response

Please see the attachment that include at the end of the file all the images of western blot  that appear in the manuscript with the molecular weight ladders.

Reviewer 2 Report

This is an interesting study that demonstrates BET inhibitor JQ1 reduces experimental glomerulosclerosis via repressing SOX9/COLIV. The authors show that JQ1 treatment diminishes glomerulosclerosis and reduces COLIV expression at both transcriptional and protein levels. Using ChIP assays, they demonstrate that JQ1 inhibits the recruitment and binding of BRD4 to the Col4a3 promoter region and thus reduces Col4a3 transcription. Using NTS-treated mice and mesangial cell culture models, the authors show that JQ1 treatment blocks SOX9 nuclear translocation and activation. In addition, SOX9 gene silencing represses COLIV production.

Several major issues need to be considered.

1.   The authors show that JQ1 reduces ECM accumulation (Figure 1) and COLIV production (Figure 2C). The Masson Trichromic staining and immunohistochemistry staining should be quantified. And it would be better to show the scale bar on the images (Figure 1,2C,6A,8 and 10C).

2.   In figure 4, the authors show that TGF-β1 induces Fibronectin and Col1a2 transcription in mesangial cells, which are inhibited by JQ1 treatment (Figure 4 A and B). What about Col4a3 transcription level? Is it induced by TGF-β1 and repressed via JQ1 treatment, which are consistent with data in figure 2, giving that JQ1 is a BET inhibitor which inhibits the recruitment and binding of BRD4?

3.   The authors show that JQ1 inhibits SOX9 transcription and nuclear translocation in NTS-treated mice and cultured mesangial cells (Figure 5,6 and 8). It would be interesting to determine the underlying mechanism through which JQ1 supresses SOX9 transcription and nuclear translocation. Does JQ1 restrain SOX9 activation via the same mechanisms as citation [84] does? Given the effects of JQ1 to inhibit the activation of transcription factor SOX9, what is the effects of SOX9 gene silencing on COLIV transcription? q-PCR experiment is needed.

4.   The authors use Red Ponceau as a loading control rather than Histone H1 or GAPDH in western blots (Figure 5B and 10B). It is not the optimal loading control in blots with nuclear proteins.

5.   In figure 6, the authors use SOX9 immunofluorescence staining to demonstrate that JQ1 inhibits SOX9 nuclear translocation induced by TGFβ1 treatment. A quantification of co-localization between SOX9 and DAPI is needed.

6.   In figure 9, the authors demonstrate that JQ1 treatment reduces phosphorylation of SMAD3 both in NTS-treated mice and cultured mesangial cells. However, what is the correlation between TGFβ1/SMAD3 pathway and SOX9/ COLIV? What are the effects JQ1 has on the proteins downstream of SMAD3?

7.   In figure 10, the authors conclude that JQ1 inhibits SOX9 nuclear translocation induced by UUO. UUO model is widely used in kidney fibrosis study, so what is the relation between UUO and glomerulosclerosis, since the title of this manuscript is Type IV collagen and SOX-9 are molecular targets of BET inhibition in experimental glomerulosclerosis? In addition, the data show that α-SMA, Fibronectin and Colâ…  are increased in UUO model (Figure 10C and E) and diminished via JQ1 treatment, what is the effects of JQ1 on COLIV both in transcription and protein levels? 

Reviewer 3 Report

Reviewer's comments,

I think that this paper is good, so I recommend accept.
